# Chemical Signaling in the Turbulent Ocean—Hide and Seek at the Kolmogorov Scale

**Erik Selander [1],*, Sam T. Fredriksson [2] and Lars Arneborg [2],***

[1]  Department of Marine Sciences, University of Gothenburg, Carl Skottsbergs g 22B,
    SE 413 19 Gothenburg, Sweden

[2]  Department of Research and Development, Swedish Meteorological and Hydrological Institute,
    Sven Källfelts Gata 15, SE 426 71 Västra Frölunda, Sweden; sam.fredriksson@smhi.se

*  Correspondence: erik.selander@marine.gu.se (E.S.); lars.arneborg@smhi.se (L.A.)

**Abstract:** Chemical cues and signals mediate resource acquisition, mate finding, and the assessment of predation risk in marine plankton. Here, we use the chemical properties of the first identified chemical cues from zooplankton together with in situ measurements of turbulent dissipation rates to calculate the effect of turbulence on the distribution of cues behind swimmers as well as steady state background concentrations in surrounding water. We further show that common zooplankton (copepods) appears to optimize mate finding by aggregating at the surface in calm conditions when turbulence do not prevent trail following. This near surface environment is characterized by anisotropic turbulence and we show, using direct numerical simulations, that chemical cues distribute more in the horizontal plane than vertically in these conditions. Zooplankton may consequently benefit from adopting specific search strategies near the surface as well as in strong stratification where similar flow fields develop. Steady state concentrations, where exudation is balanced by degradation develops in a time scale of ~5 h. We conclude that the trails behind millimeter-sized copepods can be detected in naturally occurring turbulence below the wind mixed surface layer or in the absence of strong wind. The trails, however, shorten dramatically at high turbulent dissipation rates, above ~$10^{-3}$ cm$^2$ s$^{-3}$ ($10^{-7}$ W kg$^{-1}$)

**Keywords:** Kolmogorov; turbulence; copepod; chemosensory; signaling; zooplankton

## 1. Introduction

The open ocean is a dilute environment. Organisms have to process large volumes of water to acquire resources and find mates [1]. At the same time, predation rates are high, and organisms have to trade resource acquisition against predator avoidance. These contradictory needs drive evolution of advanced sensory systems to improve detection of both resources and threats. The vast majority of marine plankton depend on hydro-mechanical and chemosensory information. Both depend on the fluid flow regime. Here, we focus on the effects of turbulence on chemical sensing and signaling at the length scale characteristic for the transition between viscous and turbulent flows, the Kolmogorov length scale.

The most numerous organisms in the sea, bacteria, experience a fully viscous flow regime. The transport of solutes is mainly governed by diffusion, and the effect of turbulence often marginal. Bacteria use simple behavioral algorithms to maneuver the chemical landscape in a way that is fairly well understood [2]. In principle, the rate of change of direction is altered in response to concentration gradients in a way that allows bacteria to navigate towards, or away from sources. Sensing and signaling in slightly larger organisms is, however, more complex. The most abundant multicellular

animals in the sea, copepods, are all in this size range, 100 μm to a few mm, and may even traverse the Kolmogorov scale when growing up from larval stages to adults (Figure 1).

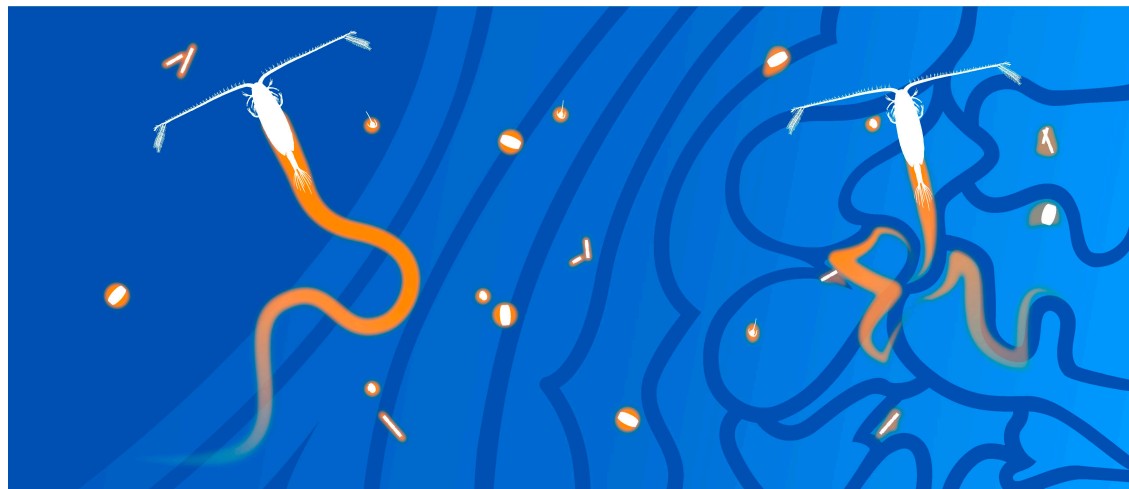

**Figure 1.** Artistic interpretation of the distribution of solutes around small (μm, e.g., bacteria and small phytoplankton) and intermediate sized particles (mm) such as motile copepods in the transition from viscous to turbulent regimes (from left to right). At micrometer scale the solute distribution is largely driven by diffusion. The chemical wake behind motile cells is eroded by diffusion faster than it forms and solute distribution is only moderately affected by turbulence. At slightly larger scales, motile organisms leave chemical trails in their wake that can be detected tens to hundreds body lengths away [3]. At higher turbulent dissipation rates however, the trails will be deformed and thinned by turbulence in a way that reduce detectability [4]. Local gradients will also be more rapidly mixed into background concentrations at high turbulence (Illustration: Jan Heuschele).

While it has long been known that copepods emit and receive chemical cues involved in mate finding [3,5], resource acquisition, and predator avoidance [6], the cueing compounds have remained largely unknown. As a consequence, fundamental parameters such as exudation rate, degradation, diffusivity, and sensitivity thresholds have been lacking. This is one of the reasons why the literature holds contrasting views on whether zooplankton can exploit chemical trails to increase encounter rates in nature, or if turbulence erodes the trace too fast (see e.g., [3,7]) in the ocean. Recently, the first signaling compounds from copepods were identified as a group of polar lipids, copepodamides [8]. Copepodamides induce defensive traits such as toxin production, bioluminescence, or colony size plasticity in prey organisms [8–10]. The exudation rate, degradation rate, and sensitivity threshold have now been established for copepodamides [8,9].

The purpose of the present study is to combine the empirical data on physical properties of cueing compounds with in-situ measurements of turbulent dissipation rates to test the theories on the effect of turbulence on sensing and signaling in the open ocean. Moreover, copepods form near surface reproductive swarms [11]. Near the surface, however, the turbulence is typically anisotropic [12–15]. We explore the effect of anisotropic turbulence on chemosensing near the surface using direct numerical simulations.

*Theoretical Background*

The physics of trail formation behind zooplankton has been thoroughly addressed by others. A simple model for the solute concentration behind a copepod in calm water is that of a point source moving at constant velocity through a diffusive medium [3]. The diffusion problem can be solved analytically [16] and the solution can be expressed as

$$C = \frac{Q}{4\pi Dz} exp\left(-\frac{Ur^2}{4Dz}\right) \tag{1}$$

where z is the along-track distance behind the copepod and r is the radial distance from the centerline of the trail, remaining parameters defined in Table 1. The same expression can be used for chemical trails behind particles falling through the water column with velocity $U$ [17]. An organism with detection limit $C^*$ can detect the trail at distance

$$z_0 = \frac{Q}{4\pi DC^*} \tag{2}$$

It is noteworthy that the detection length is independent of velocity. For a fast copepod, the trail is thinner and diffuses faster than for a slow, but the detectable trail length is the same [1].

In a turbulent environment, the trail will be stretched and thinned by turbulent straining. At scales smaller than the Kolmogorov scale, the viscous straining dominates the relative displacement of particles, and since this is the movement that removes most of the kinetic energy from the flow, there is a direct relationship between straining, viscosity, and dissipation of turbulent kinetic energy. The rate of straining, $\gamma$, scales as:

$$\gamma = \left(\frac{\varepsilon}{6\nu}\right)^{1/2} \tag{3}$$

Diffusivity tends to widen the trail whereas advective thinning due to straining tends to narrow it. A balance between diffusive widening and thinning due to strain is obtained when the trail width is about equal to the Batchelor scale, $B = (\nu D^2/\varepsilon)^{\frac{1}{4}}$ [18]. Note that this is the scale of the actual trail width, which is different from the detectable width that decays due to decreasing trail centerline concentrations. At this width, the rate of change of trail concentration scales as

$$\frac{dC}{dt} \propto -D\frac{C}{B^2} \propto -\gamma C, \tag{4}$$

i.e., the inverse of $\gamma$ is a measure for the decay time-scale of the trail concentration in a turbulent flow. Another time scale of importance for the trail is the time it takes before a trail becomes undetectable in calm water,

$$T_0 = \frac{z_0}{U}. \tag{5}$$

**Table 1.** Parameters and values used in calculations.

| Parameter | Symbol | Value | Reference |
|---|---|---|---|
| Copepod length | $L$ | 0.1 cm | |
| Copepod swimming speed | $U$ | 0.05–1 cm s$^{-1}$ | [19] |
| Copepod concentration | $n$ | 10–12 ind l$^{-1}$ | sharkweb.smhi.se |
| Emission rate of copepodamides * | $Q$ | 17 pmol ind$^{-1}$ d$^{-1}$ | [8] |
| Sensitivity threshold for copepods | $C^*$ | 1 nM | [6] and references there in |
| Sensitivity threshold for microplankton | $C^p$ | 20-80 fM | [9] |
| Degradation rate | $k$ | 0.21h$^{-1}$($C_t = C_0 e^{-0.21t}$) | [20] |
| Molecular diffusivity | $D$ | $3 \times 10^{-6}$ cm$^2$ s$^{-1}$ | [21] |
| Kinematic viscosity | $\nu$ | $10^{-2}$ cm$^2$ s$^{-1}$ | |
| Dissipation rate of turbulent kinetic energy | $\varepsilon$ | $10^{-6}$–$10^3$ cm$^2$ s$^{-3}$ | [22] |

\* Calculated from weight specific copepodamide production [8] to the copepod species (*Centropages typicus*) used in [3].

The main, non-dimensional, parameter that determines the influence of turbulence on a trail is the relation between these two time scales,

$$VJ = \gamma T_0 \tag{6}$$

which we call the *VJ* number after Visser and Jackson [4]. Visser and Jackson [4] used a statistical model of isotropic turbulence and a relatively simple trail model to show that the trail characteristics are little influenced by turbulence for *VJ* < 1 whereas the trail area, volume, and decay time scale is much smaller than in calm water for *VJ* > 100. It is noteworthy that the actual length of the trail remains as long in turbulent as in calm water, but it curls up, becomes thinner, and decays much quicker and closer to the copepod in turbulent water. In other words, the material lines are elongated in turbulent water, but the elongation causes a thinning that increases the diffusive dilution so much that the material trail length remains the same. Visser and Jacksson [4] presented the following equation for the detection time scale of a trail in turbulence

$$T = T_0 \frac{ln(VJ + 1)}{VJ} \tag{7}$$

which approaches $T_0$ for small *VJ* and 0 for large *VJ*. A similar relationship can be obtained between the trail length in calm water, $z_0$, and the detection distance behind the copepod. However, these results still remain to be validated in laboratory or in direct numerical simulations. It is also limited to isotropic turbulence and do therefore do not necessarily apply near the surface or in strongly stratified fluid.

## 2. Materials and Methods

### 2.1. Calculations of Trail Lengths and Background Concentrations

The empirical data on production, degradation, and sensitivity thresholds were collected from the literature (Table 1). Observations and samples of surface swarms of copepods were obtained by bucket-sampling the densest part of two swarms observed on the Swedish west coast (58°14′59.5″N 11°26′43.8″E and 58°52′33.5″N 11°08′43.2″E). The copepods were preserved in Lugols's solution. All copepods in 65 mL (*Centropages*) or three times 50 mL (*Calanus*) were counted and prosome length (*Calanus)* or sex (*Centropages)* was determined under a dissection microscope.

Degradation rates of copeopdamides in water is from unpublished work by Arias et al. (2020) who measured degradation rates at 19 °C in seawater.

### 2.2. Direct Numerical Simulations

Direct numerical simulations (DNS) are here used to study the turbulent flow at the sea-air interface and how the turbulence affects an inert tracer such as a chemical signal. The turbulent flow is here driven by wind stress and thermal convection. The surface shear stress condition is set to give the surface-shear based Reynolds number $Re_*^* = u_*H/\nu = 120$, where $u_* = \sqrt{\tau_0/\rho}$ is the friction velocity, $H$ is the length scale here given by the domain depth, and $\nu$ is the kinematic viscosity. $\tau_0$ is the surface shear stress and $\rho$ is the density. The water-side friction velocity can be converted to wind speed $U_{10}$ at 10 m height using first $u_{*,a} = u_* \sqrt{\rho/\rho_a}$ and then

$$\frac{U(z)}{u_{*,a}} = \kappa^{-1} \ln\left(\frac{u_{*,a}\, z}{\nu_a}\right) + 5.7 \tag{8}$$

valid for neutral conditions in the atmosphere [23]. This gives $U_{10} \approx 1.3$ m s$^{-1}$ which together with the used natural convection $Q_0 = 100$ W m$^{-2}$ give conditions similar to a summer evening with clear skies and low wind conditions, comparable to the conditions the two near surface swarming events were observed in.

The modeled $0.1 \times 0.4 \times 1$ m volume is shown in Figure 2. The stream direction and depth are denoted $x$ and $z$. $y$ is perpendicular to stream direction. The domain size is $3\pi H \times \pi H \times H$ in the $x$-, $y$- and $z$-directions where $H = 0.1204$ m. It is discretized using $1206 \times 402 \times 96$ cells. The cells are equidistant in the $x$- and $y$-directions (~0.94 mm). The mesh distribution is stretched in the $z$-direction,

where the smallest and largest cell heights, closest to the surface and bottom respectively, are ~0.098 mm and ~1.96 mm.

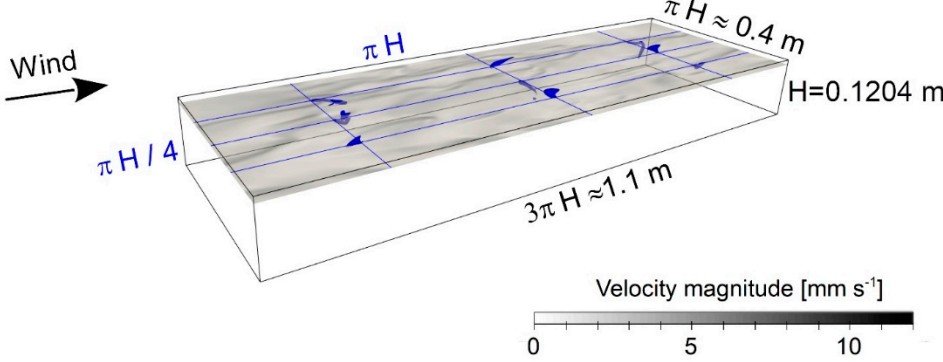

**Figure 2.** The modeled volume showing iso-concentrations of signal concentration (blue) around nine near surface point sources and the magnitude of flow velocity at the depth of the point source (0.995 cm). This plane is made transparent in order to visualize which part of the signal plumes are above and below its point of dispersal. The cross- and stream-wise positions of the point sources are given as the interceptions of the blue lines. The distances are $\pi H/4$ and $\pi H$, respectively.

The incompressible Navier–Stokes equations are solved using the standard Boussinesq approximation together with a thermal energy equation [12]. The transport equation for the passive signal is used to calculate the signal field.

$$\frac{\partial S}{\partial t} + \mathbf{U} \cdot \nabla S = D \nabla^2 S + \varnothing_S \tag{9}$$

where $S$ is the signal concentration, $t$ is time, and $\mathbf{U}$ is the fluid velocity. $D$ is the signal molecular diffusivity and $\varnothing_S$ is a spatially and temporally constant source terms added in nine positions at the same depth of 0.995 cm. The depth represents typical depth for copepod swarms and coincides with stream velocity $\bar{u} \approx 1$ mm s$^{-1}$ similar to copepod swimming speed. It is also interesting to see how anisotropic flow close to the surface influences the dispersal of the signal. The point sources are distributed evenly in both cross- and streamwise directions. This results in a $\Delta y = H \cdot \pi/4 = 0.094\ m$ and $\Delta x = H \cdot \pi = 0.378\ m$. These distances are chosen to be of the same order as the length scale $H$ and in the streamwise direction to be long enough to enable a reasonable sampling time before the different sampling sources start to interfere with each other. The diffusivity is chosen to be the same as thermal diffusivity since the small-scale gradients induced by the much smaller diffusivities of copepodamides cannot be resolved with the mesh resolution used in these simulations.

The surface boundary is assumed to be flat assuming that the surface deflection is negligible. The signal boundary condition for the surface and bottom is $\partial S/\partial z = 0$ which represent no signal exchange through these boundaries. Finally, the flow is subject to periodic (cyclic) boundary conditions in the horizontal ($x$- and $y$-) directions.

The simulations are carried out using a collocated finite volume approach with OpenFOAM, which is an open-source computational fluid dynamics tool. The computational time step $\Delta t$ is set dynamically ensuring that the Courant–Friedrich–Lewy number, $CFL = \Delta t |U|/\Delta l < 0.5$, which results in $\Delta t \approx 0.025\ s$. Here $|U|$ is the magnitude of the velocity in a cell and $\Delta l$ is the length of the cell in the velocity direction. The mesh resolution and time discretization are further discussed by [13].

Sampling of the signal concentrations is done under 60 s after 40 s of continuous seeding. The mean concentration is calculated by first superimposing an area around each of the nine seeding positions (Figure 2). The mean concentration is deducted, and a temporal average performed (Appendix A). The sampling period is short to avoid that the signal from one source interfere with adjacent sources. The values should consequently be considered qualitative rather than quantitative.

## 3. Results and Discussion

### 3.1. Mate Finding is Easier in 2 Dimensions than in 3

The most obvious way to increase encounter rate is to gather at the surface (or the bottom). This reduce the problem to two rather than three dimensions and at the same time increase concentration. Dense swarms of copepods are also observed on quiet summer days in the study area (Swedish west coast), and in e.g., freshwater copepods [11]. Among the species we observed to form swarms were *Centropages typicus*, *Acartia sp*, *Oithona* sp, *Pseudocalanus* sp, and *Calanus* sp. The swarms are usually dominated by adults of a single species (Figure 3) suggesting a reproductive role. A 65 mL sample from a *Centropages* swarm for example contained 343 males and 39 females out of which approximately half (17) had a spermatophore attached and where thus mated. Three 50 mL samples from the center of a *Calanus* swarm contained on average 1468 copepods each, corresponding to >29,000 copepods L$^{-1}$, also dominated by the large life stages although these were not sexed (Figure 3).

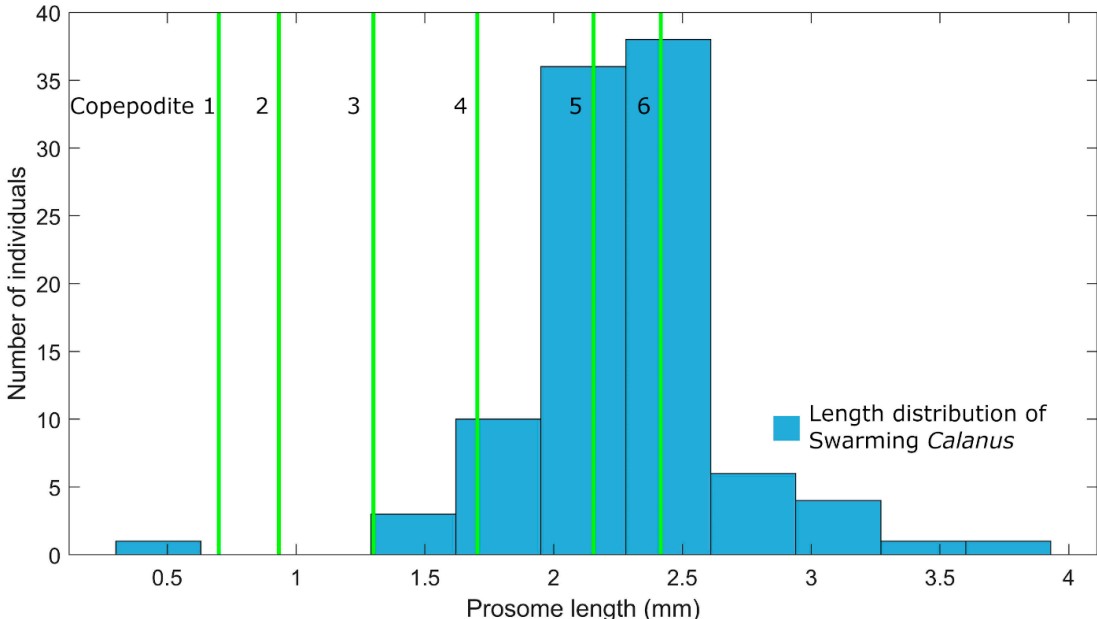

**Figure 3.** Length distribution of swarming *Calanus sp.* The green lines represent the average size of the copepodite stages (1–5 are, juveniles 6 is sexually mature adults) from monitoring data (www.sharkweb.smhi.se). The swarms are dominated by adults and late stage copepodites supporting the reproductive role of swarms. This particular swarm reached extreme densities >29,000 copepods L$^{-1}$. Most swarms are less dense.

Swarming is a powerful strategy to facilitate mate finding provided that there is a reliable cue to synchronize the ascent to the surface. Still, however, chemical information is useful for close range search and selective mating.

### 3.2. Effect of Turbulence on Chemical Trails

When applying the empirical data in Table 1 on Equation (2), copepods can easily produce trails trackable over 30–50 body-lengths in calm water, or >100 body lengths for larger copepods. Turbulence, however, deforms and dilute the chemical trails, depending on the *VJ* number as described in the introduction.

From Equations (1)–(6), one can derive the following relationship between *VJ* and $\varepsilon$

$$\varepsilon = 6\nu\left(\frac{4\pi DC^* U \, VJ}{Q}\right)^2 \tag{10}$$

For typical values of copepod swimming speed and trail length (Table 1) a value of $VJ = 1$ corresponds to a dissipation rate of $\varepsilon < 5 \times 10^{-9}$ W/kg which is calm water but not unrealistically calm below the pycnocline. A value of $VJ = 100$ corresponds to dissipation rates of $\varepsilon < 5 \times 10^{-5}$ W/kg which is a quite high dissipation rate that is seldom observed in stratified waters below the wave influenced surface layers. Faster swimming copepods have a large advantage in trail detection in turbulent water since $VJ = 1$ corresponds to $\varepsilon = 2 \times 10^{-7}$ W/kg for a fast swimming copepod ($U = 0.01$ m/s) and $\varepsilon = 5 \times 10^{-10}$ W/kg for a slow copepod ($U = 0.0005$ m/s).

The value of $z/z_0 = UT/z_0$ (trail detection distance divided by trail length in the absence of turbulence, $T$ is the detection time scale given in Equation (7)) is shown in Figure 4 for a tidal stratified flow over the Oslo fjord sill using the typical values from Table 1 and dissipation rates estimated from observations with a semi free-falling microstructure shear probe [24]. Seuront and colleagues [7] suggest that trail following is unlikely in the upper ocean. Trails are indeed shortened by >80% in the rather extreme turbulence behind the sill, but there are also calm layers where the trails are only moderately influenced by turbulence.

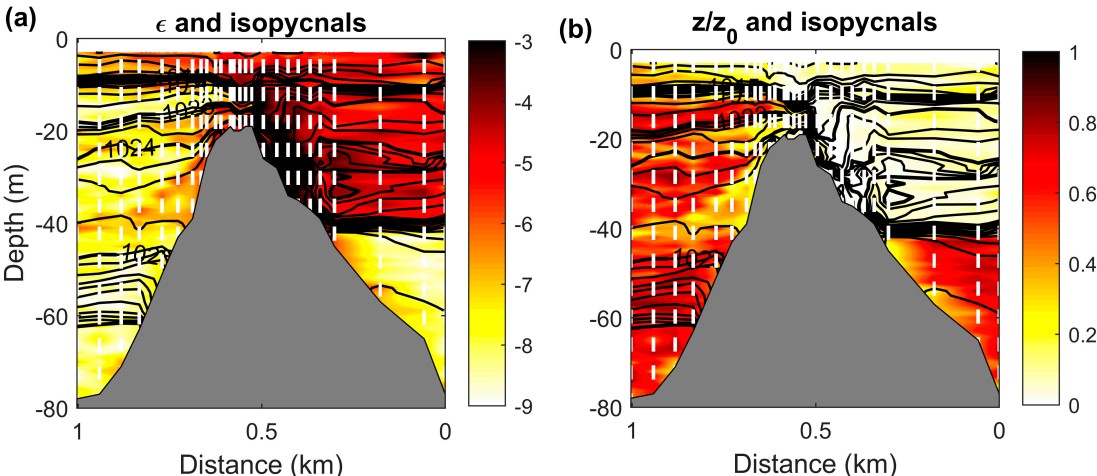

**Figure 4.** Transects over the Drøbak sill in Oslo fjord of (**a**) dissipation rates of turbulent kinetic energy ($\log_{10}$(W/kg)) [24] and (**b**) trail detection distance relative to that in calm water as functions of distance over the sill and depth for a copepods with swimming speed 0.0015 m/s and detection limit 1 nM. Black contours are lines of constant density (kg m$^{-3}$), white hatched lines indicate the sampling locations. The transect is performed during spring tide with a strong current from left to right pushing dense water over the sill that cascades down on the lee side and jumps back up again creating strong turbulence.

### 3.3. Background Concentration

Even for strong turbulence, the constant supply of copepodamides from copepods will cause an increase in background concentration, which when sufficiently large will be balanced by the decay rate of the solute. These background concentrations reach bioactive concentrations in the upper ocean that trigger defensive traits such as harmful algal toxin production [9]. Since zooplankton migrate to deeper waters during daytime to avoid visual predators, the concentration will oscillate. To understand what effective concentration that results up we need to calculate both the time scale, and the equilibrium concentration. The equilibrium between supply and decay can be written as

$$nQ = k\langle C \rangle \tag{11}$$

where <C> is the average concentration at equilibrium, which can be written as

$$\langle C \rangle = \frac{nQ}{k} \tag{12}$$

The time scale to reach this equilibrium is $k^{-1}$. For typical values (Table 1) the equilibrium concentration is ~30 pM which is reached in a time scale of ~5 h. This is an order of magnitude higher than the highest concentrations found in situ [9] which suggest that copepods do not reside sufficiently long in the same water package, or that other loss factors such as vertical mixing and sedimentation removes copepodamides. Alternatively, the production rate of copepodamides reported for *Calanus* sp. [8] may not be representative for other copepods and temperatures.

### 3.4. Effect of Near Surface Anisotropic Turbulence on Trail Formation

Figure 5 shows snapshots of the stream-wise and vertical velocity. The horizontal velocity is higher than the vertical at this depth and flow situation. The variability is also higher in the horizontal direction, $u_{rms}/w_{rms} \approx 2$. Similar variability ratios are found in channel flow experiments for wind-driven turbulence [14]. This ratio increases with increasing wind stress and vicinity to the surface. This anisotropy distributes signals more horizontally than vertically (Figure 6, Supplementary Video S1). The vertical spread of the plume is, beside turbulence and diffusion, enhanced by the falling sheets of fluid. The mean concentration field is not fully converged due to the limited sampling time which resulted from computational resource limitations and too avoid interference from adjacent sources. However, it is possible to conclude that a vertical search pattern will be more rewarding than a horizontal due to the anisotropy at this depth. The resolution of the model sets a limit on how low signal molecular diffusivities that can be resolved, which gives that the diffusivity used is representative for heat rather than solutes and therefore too high. The signal spread is determined by the diffusion and advection, where a high molecular diffusivity leads to more pronounced diffusion compared to a low diffusivity. Here, advection drives the anisotropic spread whereas the diffusion works towards a more uniform distribution. The anisotropic spread may consequently be slightly more pronounced for copepodamides due to their lower molecular diffusivity. The model can therefore be seen as a conservative approach regarding the anisotropic spread. The copepods are fixed in these simulations, which means that the trail is formed by an intermittent flow, however with a mean flow of 1 mm s$^{-1}$, past the copepod. In reality the trails will be more continuous due to a more constant velocity of the copepods relative to surrounding water. Supplementary Video S1 shows an animation of the spread around point sources.

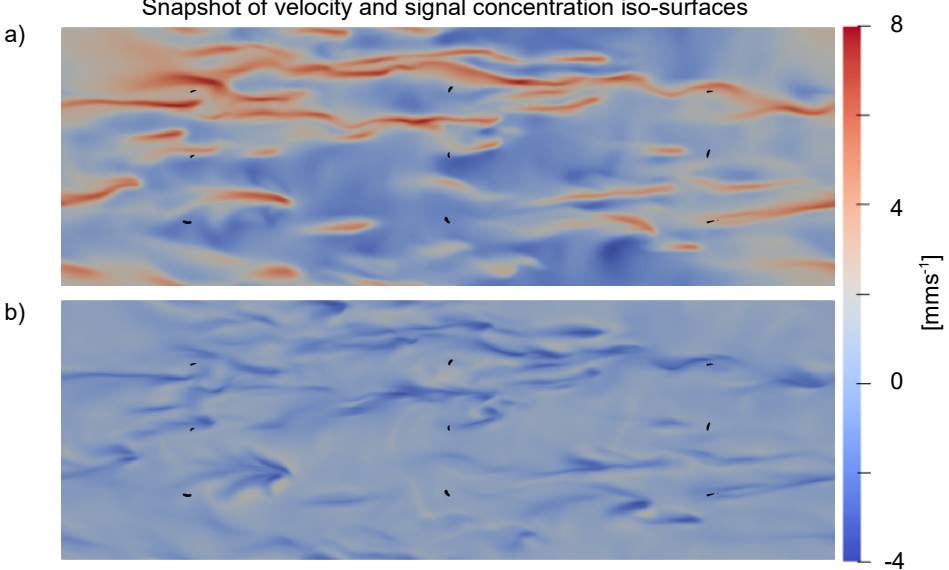

**Figure 5.** Flow velocity in the plane where the signal is seeded to the domain. Note the same signal scale close to the maximum scale for the stream wise velocity. The nine positions can be seen as the black dots (iso-surfaces of signal concentrations). (**a**) Snapshot ($t = t_0 + 40$ s) of streamwise velocity. (**b**) Snapshot ($t = t_0 + 40$ s) of vertical velocity.

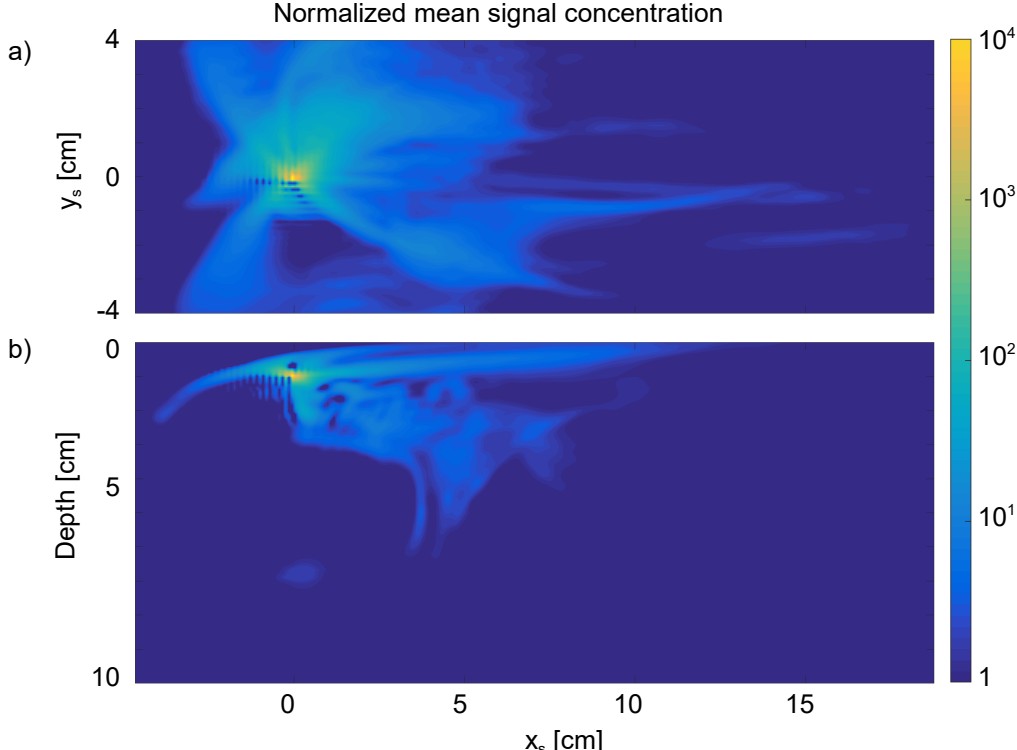

**Figure 6.** Normalized mean signal concentration. Values below one (dark blue) means that the concentration is less than the total average signal concentration. $x_s = y_s = 0$ and depth = 1 cm define the signal source position. (**a**) Horizontal plane at the seeding depth. (**b**) Vertical plane through the seeding position.

Surfactants, surface-active chemical agents at the ocean-atmosphere of water–air interface, may influence the turbulence in the vicinity of the surface [15,25] and might also attract amphiphilic compounds like the copepodamides and hence serve as a sink for signaling compounds. In addition, the surfactant typically decreases turbulence close to the surface as compared to a clean surface [15].

## 4. Conclusions

We apply new empirical data on signal substance properties on existing theories on the shortening of trail detection distance by turbulence. This reveals that trails of small copepods are affected by turbulence at quite low dissipation rates of turbulent kinetic energy, whereas the trails of large copepods are unaffected at moderate dissipation rates, which are frequently occurring in calm weather or below the wind mixed layer of stratified natural water. Copepods appear to be aware and maximize encounter rates by forming near surface swarms on quiet days when turbulence levels allow trail following. The anisotropic turbulence in this part of the ocean leads to more horizontal than vertical distribution of cues. Copepods may consequently benefit from adopting more vertical oriented search strategies to encounter chemical trails in this part of the ocean. Moreover, the timescale needed to reach steady state concentrations where exudation is balanced by degradation is in the order of 5 h, which suggests that cue concentration in surface water will oscillate with zooplankton diurnal migrations in a sinusoidal way.

**Supplementary Materials:** The following are available online at http://www.mdpi.com/2311-5521/5/2/54/s1, Video S1: Solute distribution around point sources in anisotropic turbulence.

**Author Contributions:** Conceptualization, E.S., S.T.F. and L.A.; methodology, E.S., S.T.F. and L.A.; writing—original draft preparation, E.S., S.T.F. and L.A.; writing—review and editing, E.S., S.T.F. and L.A.; visualization, S.T.F. and L.A.; funding acquisition, E.S., S.T.F. and L.A. All authors have read and agreed to the published version of the manuscript.

**Funding:** This research was funded by Swedish Research Council VR, grant numbers 2015-05491 and 2019-05238 (to ES). CoCliME which is part of ERA4CS, an ERA-NET initiated by JPI Climate, and funded by EPA (IE), ANR (FR), DLR (DE), UEFISCDI (RO), RCN (NO) and FORMAS (SE), with co-funding by the European Union (Grant 690462).

**Conflicts of Interest:** The authors declare no conflict of interest. The funders had no role in the design of the study; in the collection, analyses, or interpretation of data; in the writing of the manuscript, or in the decision to publish the results.

## Appendix A  Calculation of Normalized Mean Signal Concentration

The spatial and temporal averaged scalar concentrations are calculated as follows. The scalar is added to the domain from time $t_0$ until $t = t_0 + 100$ s. The sampling is done between at $t_1 = t_0 + 40$ s to $t_2 = t_0 + 100$ s with 2.5 s between each realization (snapshot). There are nine positions and for each time realization first the differential concentration is calculated as

$$\Delta S^m_{ijk,t} = S^m_{ijk} - S_t \tag{A1}$$

where $m$ is the position number, $ijk$ are the indices in the *x*-, *j*-, and *z*-direction for that subdomain, $t$ is the time of the realization and $S_t$ is the total amount in the whole domain during that realization. Then the spatial and temporal average for a scalar domain is found by

$$\Delta \bar{\bar{S}}_{ijk} = \frac{\sum_{m=1}^{9} \sum_{t=t_1}^{t_2} \Delta S^m_{ijk,t}}{t \cdot m} \tag{A2}$$

that in turn are normalized using the total amount of scalar at the last realization $S_{t_2}$ as

$$\Delta \widetilde{\bar{\bar{S}}}_{ijk} = \Delta \bar{\bar{S}}_{ijk} / S_{t_2} \tag{A3}$$

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
