# Peer review of "Chemical Signaling in the Turbulent Ocean—Hide and Seek at the Kolmogorov Scale"

_fluids, doi:10.3390/fluids5020054_

Round 1

Reviewer 1 Report

Referee report on the paper: “Chemical signaling in the turbulent ocean -Hide and seek at the Kolmogorov scale” by Selander et al.

This is an interesting paper discussing the detectability of chemical signals trailing moving organisms in a turbulent aquatic environment. The presentation is concise and the paper is well written, although a significant part is a review of known results. This review part fits, however, well into the main part which contains a sufficient amount of new material to warrant publication. The video attached to the manuscript offers useful supporting information. In principle the paper can be published in its present form as far as I am concerned. There are however a few comments the authors should consider.

Figure 4 contains a large number of small white vertical lines. They seem systematic, what are these indicating? Probably it is of minor importance, but a line of explanation would be good.

The conclusion section 4 is very short and should be expanded.

There is an opinion that the authors might consider: at first reading, I started imagining how I would have approached the problem myself. There are well established results for elongations of a filament in the literature, e.g., Batchelor, ‘‘The effect of homogeneous turbulence on material lines and surfaces,’’ Proc. R. Soc. London, Ser. A 213, 349 (1952) is the standard reference, but there others. At late times, the elongation of a filament is exponential. Apart from a numerical constant, the time scale for the elongation is the Kolmogorov time scale squareroot(nu/epsilon), with “nu” being viscosity and “epsilon” the specific energy dissipation rate. The standard analysis refers to a line element, but is reasonable to assume the result applies for a fine length element. The analytical expression assumes the length of the filament to be in the inertial subrange. There are not many studies of this problem (not that I know of) but some numerical simulations support this result, Krane et al. “Concentration fluctuations in two-dimensional turbulence” Europhys. Lett., 36, 99 (1996). Since the flow is incompressible, the volume of the filament as the one shown trailing plankton in left part of figure 1 is conserved as long as its boundary is well defined, i.e. not too much eroded by turbulence or molecular diffusion. The length L of the filament is given in the paper, but does not really enter the analysis here, all we need to assume is that it exists. With the volume Pi x L x D conserved, the diameter D of the filament must decrease from its initial value (the one close to plankton) as L increases, until D reaches a scale size where the concentration is no longer detectable.  Given a priori information, the time it takes should be calculable and a corresponding overall filament length can be estimated. In a real flow, the contraction will be intermittent, strong some places, weaker other, but as an estimate I think this should work. These are only some scattered remarks for the authors’ considerations, but I believe some comments on the elongation of line elements in a turbulent flow will be in order, for instance in the conclusion section.

I have no comments on the English presentation.

Author Response

Dear Referee 1, we would like to take the opportunity to thank you for your time and efforts reviewing our manuscript. Below you can see, in red, our response to the specific comments. Best wishes /Selander, Fredrikson, Arneborg

Referee report on the paper: “Chemical signaling in the turbulent ocean -Hide and seek at the Kolmogorov scale” by Selander et al.

This is an interesting paper discussing the detectability of chemical signals trailing moving organisms in a turbulent aquatic environment. The presentation is concise and the paper is well written, although a significant part is a review of known results. This review part fits, however, well into the main part which contains a sufficient amount of new material to warrant publication. The video attached to the manuscript offers useful supporting information. In principle the paper can be published in its present form as far as I am concerned. There are however a few comments the authors should consider.

Figure 4 contains a large number of small white vertical lines. They seem systematic, what are these indicating? Probably it is of minor importance, but a line of explanation would be good.

Thank you for pointing this out, the lines refers to the stations where vertical profiles where obtained. Now explained in the figure legend.

The conclusion section 4 is very short and should be expanded.

There is an opinion that the authors might consider: at first reading, I started imagining how I would have approached the problem myself. There are well established results for elongations of a filament in the literature, e.g., Batchelor, ‘‘The effect of homogeneous turbulence on material lines and surfaces,’’ Proc. R. Soc. London, Ser. A 213, 349 (1952) is the standard reference, but there others. At late times, the elongation of a filament is exponential. Apart from a numerical constant, the time scale for the elongation is the Kolmogorov time scale squareroot(nu/epsilon), with “nu” being viscosity and “epsilon” the specific energy dissipation rate. The standard analysis refers to a line element, but is reasonable to assume the result applies for a fine length element. The analytical expression assumes the length of the filament to be in the inertial subrange. There are not many studies of this problem (not that I know of) but some numerical simulations support this result, Krane et al. “Concentration fluctuations in two-dimensional turbulence” Europhys. Lett., 36, 99 (1996). Since the flow is incompressible, the volume of the filament as the one shown trailing plankton in left part of figure 1 is conserved as long as its boundary is well defined, i.e. not too much eroded by turbulence or molecular diffusion. The length L of the filament is given in the paper, but does not really enter the analysis here, all we need to assume is that it exists. With the volume Pi x L x D conserved, the diameter D of the filament must decrease from its initial value (the one close to plankton) as L increases, until D reaches a scale size where the concentration is no longer detectable.  Given a priori information, the time it takes should be calculable and a corresponding overall filament length can be estimated. In a real flow, the contraction will be intermittent, strong some places, weaker other, but as an estimate I think this should work. These are only some scattered remarks for the authors’ considerations, but I believe some comments on the elongation of line elements in a turbulent flow will be in order, for instance in the conclusion section.

The thinning is part of the straining, and this has now been made more clear in the text. “..and thinned” has been added to the first sentence in the paragraph including Eq. 3. “...advective thinning due to …” has also been added to the sentence after Eq. (3).  We are considering scales smaller than the Kolmogorov scale, where the thinning will be important until it is balanced by widening and dilution due to diffusion. This is well described in Batchelor (1959) which should have been, and is now, cited. We have also added “In other words, the material lines are elongated in turbulent water, but the elongation causes a thinning that increases the diffusive dilution so much that the material trail length remains the same” before Equation (7). The theory is mainly a review and therefore it is mainly the application on empirical data that is included in the conclusion.

In addition the manuscript has been reworked in several other places suggested by the other two referees, a version with track changes on is provided here for clarity.

Reviewer 2 Report

This article presents an interesting discussion on how copepods leave chemical traces in ocean. The article provides a good review of the literature and physics of the problem. It is particularly interesting to think that the anisotropy of the ocean currents preferentially dispersed their trail horizontally and this reduces the problem of tracking a signal to two-dimensions rather than three. 

However, I do not think that the article presents sufficient evidence to reach the conclusions presented. Specifically, I have the following points:

1) The authors begin with the premise that the copepod trails will remain stratified but appear to design the DNS to support that pre-made conclusion. If you design a simulation with flow properties that favour horizontal diffusion, then you can presume the results will indicate this. In fact, I would argue that the DNS results only mildly support this conclusion as the mean concentration fields in figure 6 do not look converged. Some key information such as the mesh resolution and time discretization of the simulation are also missing.

2) The authors also posit in the conclusion that the trails drastically shorten at high dissipation rates, however, I don't believe that they actually tested more than one dissipation rate.

3) To be fair, the authors do use describe some basic calculations to support these points in section 3.2, presumedly using the equations presented in the introduction, however, not enough steps of the calculations are provided to properly follow.

4) Finally, I am not sure what is the point of section 3.3 Background concentration. The authors calculate the expected background concentration then present three reasons why this is incorrect. What does this add to the discussion?

Overall, the article is well written and does provide an interesting contribution to the topic but I recommend resubmitting after a careful re-writing.

Author Response

Dear referee #2, we would like to take the opportunity to thank you for you r time and effort reviewing the manuscript. Detailed response to individual comments below in red

This article presents an interesting discussion on how copepods leave chemical traces in ocean. The article provides a good review of the literature and physics of the problem. It is particularly interesting to think that the anisotropy of the ocean currents preferentially dispersed their trail horizontally and this reduces the problem of tracking a signal to two-dimensions rather than three. 

However, I do not think that the article presents sufficient evidence to reach the conclusions presented. Specifically, I have the following points:

1) The authors begin with the premise that the copepod trails will remain stratified but appear to design the DNS to support that pre-made conclusion. If you design a simulation with flow properties that favour horizontal diffusion, then you can presume the results will indicate this. In fact, I would argue that the DNS results only mildly support this conclusion as the mean concentration fields in figure 6 do not look converged. Some key information such as the mesh resolution and time discretization of the simulation are also missing.

The simulation as such (low wind with thermal convection) is not designed to favour horizontal dispersal. It is, however, a common flow situation (and depth) for swarming copepods and therefore interesting to study.  To further enhance this “The horizontal velocity is higher than the vertical. “ is changed into “The horizontal velocity is higher than the vertical at this depth and flow situation.”    

We agree that the concentration field is not fully converged. It is stated in Section 2.2. that “The sampling period is short to avoid that sources interfere with adjacent sources. The values should consequently be considered qualitative rather than quantitative.” And we modified the discussion to:  “The mean concentration field is not fully converged due to the limited sampling time which resulted from computational resource limitations and too avoid interference from adjacent sources. Yet it is possible to conclude that a vertical search pattern will be more rewarding than a horizontal due to the anisotropy at this depth.” 

We have expanded the description of the DNS in materials and methods and also modified figure 2 to include more details. Please see attached file with track changes activated for details.  Finally we added an appendix describing how the mean concentration fields in Figure 6 are calculated. 

2) The authors also posit in the conclusion that the trails drastically shorten at high dissipation rates, however, I don't believe that they actually tested more than one dissipation rate.

The reduction in detection distance is not a result of this study and therefore removed. Instead we include the result of this work that is the application of empirical data on earlier theoretical models: “We apply new empirical data on copepod signal substance properties on existing theories on the shortening of trail detection distance by turbulence. This reveals that trails of small copepods are affected by turbulence at quite low dissipation rates of turbulent kinetic energy, whereas the trails of large copepods are unaffected at moderate dissipation rates, which are frequently occurring in calm weather or below the wind mixed layer of stratified natural water. “

3) To be fair, the authors do use describe some basic calculations to support these points in section 3.2, presumedly using the equations presented in the introduction, however, not enough steps of the calculations are provided to properly follow.

Thank you for pointing this out, we have included an expression for the relationship between VJ and eps (Eq 1O) and additional references to the equations in the introduction.

4) Finally, I am not sure what is the point of section 3.3 Background concentration. The authors calculate the expected background concentration then present three reasons why this is incorrect. What does this add to the discussion?

The background concentrations reached, and the time scale is mainly of ecological significance. The emitting zooplankton migrate to deeper waters each day to avoid visual predators. The compound concentration will as a consequence oscillate. Prey organisms react to the amount of compound in the water, and both the time scale and resulting background concentrations are key to  understand what effective concentrations will result. We have added a couple of sentences to motivate this calculation:

These background concentrations reach bioactive concentrations in the upper ocean that trigger defensive traits such as harmful algal toxin production [9]. Since zooplankton migrate to deeper waters during daytime to avoid visual predators, the concentration will oscillate. To understand what effective concentration that results up we need to calculate both the time scale, and the equilibrium concentration.”

Overall, the article is well written and does provide an interesting contribution to the topic but I recommend resubmitting after a careful re-writing.

Reviewer 3 Report

This paper might be of interest to a wide range of audience. However, the writing is too confusing. It has observations from different places and channel flow turbulence simulations with unclear setup stated. It does not convey a coherent and focused story. I would encourage resubmission, if they can make their results or potential more results into a coherent, clear and focused story.
1. Introduction needs some rewriting: too many equations; and does not state clearly the knowledge gap, your specific scientific goal, the motivation and outline of the work. So I suggest you rewrite the introduction carefully.

2. Eq. (9): There are both capital letter S and lower case s in the equation. Please be consistent.

3. Line 150: “in nine positions at the same depth of approximately 1 cm.”
Please be clear about the location of these nine positions. Maybe you can add these positions in Figure 2. Is it the blue dots in Figure 2. If yes, you should say this explicity in the caption of Fig. 2.

4. In section 2, state clearly of your choice of D in Eq. (9). Also state clearly the rationale of your seeding position.

5. Fig3: Please state clearly the ocean region or the source of the data (describe in detail).
Line 178: what is the latitude and longitude of this laboratory? I am confused, you also mention “Swedish west coast” on line 119. Are the laboratory from line 178 is in the Swedish west coast? Also two few details of the data is written here (e.g. how is the data obtained and what are measured).

6. Fig.5 and 6: How does your result and conclusion here are sensitive to the choice of D in Eq. (9) and the seeding positions.

Author Response

Dear referee #3, we would like to take the opportunity to thank you for your time and effort reviewing the manuscript. Detailed response to the individual comments below in red:

This paper might be of interest to a wide range of audience. However, the writing is too confusing. It has observations from different places and channel flow turbulence simulations with unclear setup stated. It does not convey a coherent and focused story. I would encourage resubmission, if they can make their results or potential more results into a coherent, clear and focused story.

The setup of channel flow simulation is now more precisely described in section 2.2. By (some formatting does not come across very well in this online format, we suggest to view the attached word document with "track changes" activated for full details)

The modeled 0.1*0.4*1 m volume is shown in Figure 2. The stream direction and depth are denoted  x and z, y is perpendicular to stream direction. The domain size is 3πH*πH*πH  in the x, y, and z directions where  0.1204 m. It is discretized using 1206 x 402 x 96 cells. The cells are equidistant in the x- and y-directions (~0.94 mm). The mesh distribution is stretched in the z-direction, where the smallest and largest cell heights, closest to the surface and bottom respectively, are ~0.098 mm and ~1.96 mm”.

and

“D is the signal molecular diffusivity and øs is a spatially and temporally constant source terms added in nine positions at the same depth of 0.995 cm. The depth represents typical depth for copepod swarms and coincides with stream velocity u=1 mms-1similar to copepod swimming speed. It is also interesting to see how anisotropic flow close to the surface influences the dispersal of the signal. The point sources are distributed evenly in both cross- and streamwise directions. This results in Δy=H*π/4=0.094 m and Δx=H*π=0.378 m. These distances are chosen to be of the same order as the length scale  and in the streamwise direction to be long enough to enable a reasonable sampling time before the different sampling sources start to interfere with each other. “

and

“The computational time step Δs is set dynamically ensuring that the Courant-Friedrich-Lewy number, CFL=ΔtIUI/Δl< 0.5, which results in Δt=0.025 s Here IUI is the magnitude of the velocity in a cell and is the length of the cell in the velocity direction. The mesh resolution and time discretization are further discussed by [12].”

and the figure 2 has been updated and with an extended caption

“The modeled volume showing iso-concentrations of signal concentration (blue) around nine near surface point sources and the magnitude of flow velocity at the depth of the point source (0.995 cm). This plane is made transparent in order to visualize which part of the signal plumes are above and below its point of dispersal. The cross- and stream-wise positions of the point sources are given as the interceptions of the blue lines. The distances are πH/4 and πH, respectively.”

, see also point 3 below. 

  1. Introduction needs some rewriting: too many equations; and does not state clearly the knowledge gap, your specific scientific goal, the motivation and outline of the work. So I suggest you rewrite the introduction carefully.

Thank you for this feedback, we have rearranged the introduction and confined the equations to a section called "theoretical background". We find the equations crucial to describe the current knowledge, and also refer to them later in the text. In addition we have stressed the knowledge gap and specific scientific goals more explicitly in the last paragraph before the theoretical background

Pls see the attached file with track changes activated to see details of the revision.

  1. Eq. (9): There are both capital letter S and lower case s in the equation. Please be consistent. 

It is subscript capital S.

  1. Line 150: “in nine positions at the same depth of approximately 1 cm.” 

Please see the general answer regarding unclear DNS setup. 

Please be clear about the location of these nine positions. Maybe you can add these positions in Figure 2. Is it the blue dots in Figure 2. If yes, you should say this explicity in the caption of Fig. 2.

You cannot see the specific positions in Fig. 2 since they are “inside” the concentration isosurface. The figure has now been updated with lines where the inception indicates the specific positions. 

  1. In section 2, state clearly of your choice of D in Eq. (9). Also state clearly the rationale of your seeding position.

Seeding position is further explained in the general answer regarding unclear DNS setup.

The choice of D is discussed under point 6 below.

  1. Fig3: Please state clearly the ocean region or the source of the data (describe in detail). 

Line 178: what is the latitude and longitude of this laboratory? I am confused, you also mention “Swedish west coast” on line 119. Are the laboratory from line 178 is in the Swedish west coast? Also two few details of the data is written here (e.g. how is the data obtained and what are measured).

Thank you for noticing this, we have added the positions of both swarms to the materials and methods section together with some details on the measurements:

Observations and samples of surface swarms of copepods were obtained by bucket-sampling the densest part of two swarms observed on the Swedish west coast (58°14'59.5"N 11°26'43.8"E and 58°52'33.5"N 11°08'43.2"E). The copepods were preserved in Lugols’s solution. All copepods in 65 ml (Centropages) or three times 50 ml (Calanus) were counted and prosome length (Calanus) or sex (Centropages) was determined under a dissection microscope.

  1. Fig.5 and 6: How does your result and conclusion here are sensitive to the choice of D in Eq. (9) and the seeding positions.

The text in section 3.4 is changed to:

“The resolution of the model sets a limit on how low signal molecular diffusivities that can be resolved, which gives that the diffusivity used is representative for heat rather than solutes and therefore too high. The signal spread is determined by the diffusion and advection, where a high molecular diffusivity leads to more pronounced diffusion compared to a low diffusivity. Here, advection drives the anisotropic spread whereas the diffusion works towards a more uniform distribution. The anisotropic spread may consequently be slightly more pronounced for copepodamides due to their lower molecular diffusivity. The model can therefore be seen as a conservative approach regarding the anisotropic spread.”

Round 2

Reviewer 2 Report

I am satisfied that the authors have adequately addressed all my comments and that the article is of high quality now to be a good contribution to this journal.

Author Response

Thank you for your time and effort reviewing the manuscript, we found your comments very helpful. Best wishes, Erik Selander on behalf of all the authors

Reviewer 3 Report

I see that the authors have made significant efforts improving the presentation of their work.

Here are some minor comments:

(1)Line 64: “have not been lacking “ or “have been lacking“?

(2)Line 79: “1.1” should be “1” here.

(3)Line 80: Move Fig.2 to line 163,   so that it can be right above the figure caption.

Author Response

Thank you again for your useful comments that helped us improve the manuscript. 

We changed the three additional minor comments in line with your suggestions with your suggestions except comment no 2, see below for details.

(1)Line 64: “have not been lacking “ or “have been lacking“?

changed to "have been lacking"

(2)Line 79: “1.1” should be “1” here.

The "1.1" is a result of the heading format in Fluids, the "Introduction" is numbered 1 and since  "Theoretical background" is a subheading within the introduction it cannot have the same number. 

(3)Line 80: Move Fig.2 to line 163,   so that it can be right above the figure caption.

We moved the figure to line 163